# How Does a Regulatory Minority Shareholder Influence the ESG Performance? A Quasi-Natural Experiment

**Di Song** [1], **Canyu Xu** [2,*] , **Zewei Fu** [1] **and Chao Yang** [3]

1 Business School, China University of Political Science and Law, Beijing 100088, China; cu202018@cupl.edu.cn (D.S.); zeweifu@163.com (Z.F.)
2 Business School, East China University of Science and Technology, Shanghai 200237, China
3 School of Accountancy, Beijing Wuzi University, Beijing 101149, China; ycbm132@126.com
* Correspondence: lucky_0124@126.com

**Abstract:** Based on China's newly established Securities Investor Services Center (CSISC), a minority shareholder protection mechanism, we investigated how the CSISC shareholder influences the ESG performance of listed companies. Using a difference-in-differences analysis for a sample of Chinese listed companies during 2013–2017, we found that the pilot reform of CSISC shareholding has a positive influence on the ESG performance of listed companies. We also found that this effect exists in large companies and in companies in non-high-polluting industries. Besides, analysts' attention, external auditing quality, institutional shareholding, and highly-developed market intermediary and legal systems can strengthen the effect of CSISC shareholding on corporate ESG performance. Our findings inspire regulators in emerging markets to establish suitable mechanisms to protect minority shareholder rights in the long run.

**Keywords:** China Securities Investor Services Center; minority shareholder protection; ESG performance; supervision capability; information transparency

## 1. Introduction

Investor protection is prominent for the healthy development of the capital market [1–4]. However, the protection of minority shareholders is still poor in many emerging economies. In China, the shareholder structure of listed companies in the A-share market presents a highly concentrated feature, while most of the accounts in the market are held by individual investors [5]. Minority shareholders lack the necessary professional knowledge and rational trading ability, leading to a weak ability to protect legal rights. Hence, they face greater investment losses due to issuers' fraudulent issuance, internal trading, false statements, and market manipulation [6]. Therefore, the development of an effective investor protection mechanism is urgent. To improve investor protection, the China Securities Investor Services Center (CSISC) was established by the China Securities Regulatory Commission (CSRC), which aims to exercise various shareholder rights to protect minority shareholders' benefits.

The existing research exploring the impact of CSISC on the capital market is relatively limited, mainly focusing on the economic consequences of its establishment. CSISC shareholding encourages minority shareholders' activism and plays a role in protecting the rights and interests of minority shareholders; therefore, the pilot reform of CSISC shareholding can significantly reduce the stock price crash risks, promote the smooth operation of the financial market [7], reduce earnings management, and improve earnings quality [5]. Based on the existing literature, this paper further focused on how CSISC can improve the long-term sustainability performance of companies, which means the impact on ESG performance. We found that the pilot reform of CSISC shareholding has a positive influence on the ESG performance of listed companies. The CSISC, as a minority shareholder, has a positive influence on the ESG through increased supervision capability and information transparency.

We focused on the impact of ESG performance mainly because minority shareholders have insufficient ability to exercise individual interests [6]. In emerging economies with concentrated ownership structures, large shareholders, especially controlling shareholders, are likely to pose a negative influence in monitoring a firm's opportunistic behaviors and, in turn, increase private short-term interest. Compared with large shareholders, benefiting from the short-term zero-sum game is hard for minority shareholders [8], and they pay more attention to the long-term and sustainability development of companies [9,10]. Under CSISC shareholding, some of the short-term benefits obtained by large shareholders are redirected to projects that benefit all shareholders in the long run, such as ESG. Furthermore, we believe it is necessary to investigate how CSISC ensures long-term sustainable development, which is the ESG performance of listed companies, by exercising minority shareholder rights.

To analyze the relationship between CSISC shareholding and ESG performance, we constructed a difference-in-differences model. Due to the CSISC only buying and holding 100 shares of each listed company in Shanghai, Guangdong (excluding Shenzhen), and Hunan provinces during the pilot period in 2016, we use listed firms in the pilot regions as treatment firms and listed firms in other provinces as control firms. We performed a difference-in-differences (DID) approach and found that after the CSISC shareholding pilot reform, the ESG score in pilot regions increased, which confirmed our Hypothesis 1 that the pilot reform of CSISC shareholding has a positive influence on the ESG performance of listed companies. To validate our DID research design, we performed a parallel trend analysis, finding that the differences in ESG performance in the pilot period are not attributable to the trending differences between the treatment and control groups. Our DID research design is valid. We also conducted a placebo test, propensity score matching, controlling time and regional effects, and changing the control group to confirm the robustness of our results.

Furthermore, from heterogeneity analysis, CSISC shareholding has a positive influence on large companies and companies in non-high-polluting industries. From cross-sectional analysis, analysts' attention, external auditing quality, institutional shareholding, and highly-developed market intermediary and legal system can strengthen the effect of CSISC shareholding on corporate ESG performance. The mechanism analysis shows that CSISC shareholding leads to a positive influence on ESG through increased supervision capability and information transparency. Figure 1 shows the research framework of this paper.

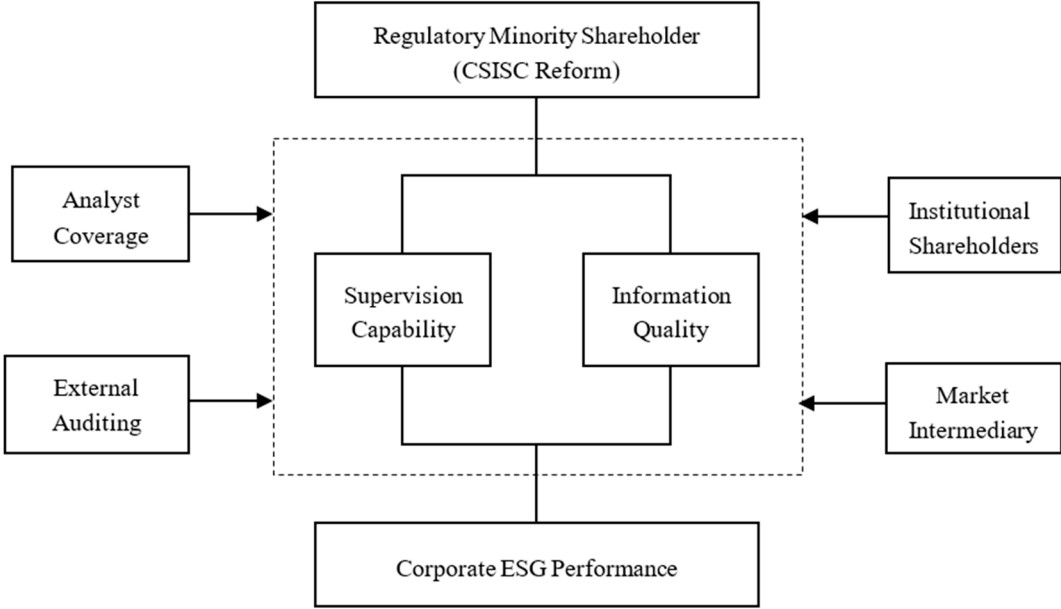

**Figure 1.** Research Framework.

Our study makes several contributions. Firstly, it contributes to the literature on minority shareholder protection and investigates the economic consequences of CSISC shareholding. Current research mainly focuses on the impact of CSISC shareholding on the short-term economic behavior of companies, including stock price crash risks [7] and earnings management [5]. However, only by improving the long-term value and focusing on the long-term sustainable development of listed companies can the investment value of minority shareholders be truly enhanced and their interests protected. Our findings can inspire regulators in emerging markets to establish suitable mechanisms to protect minority shareholder rights in the long run.

Secondly, now that the ESG score is a popular indicator for investors to select stocks, companies need to improve their ESG performance. Existing studies mainly focused on the economic consequences of the ESG [11–14] and rarely investigated how to improve ESG performance, especially from the perspective of a minority shareholder. This paper then adds to the research literature on minority shareholder protection.

Thirdly, previous studies on the relationship between minority shareholders' activism and the long-term development of listed companies were primarily conducted in the context of developed markets. Our analysis also enhances the understanding of investor protection and ESG activities in China because lowering the high ownership concentration and protecting minority investors from improving market efficiency should be the key to solving agency problems in the Chinese capital market and for all emerging economies.

The remainder of this paper proceeds as follows: Section 2 introduces the institutional background of the CSISC and the literature review. Section 3 develops our hypothesis. Section 4 describes the sample and research design. Section 5 presents our main empirical results. Section 6 concludes the paper.

## 2. Institutional Background and Literature Review

### 2.1. CSISC Shareholding

LaPorta et al. [15] proposed that investor protection in common-law countries is stronger than that in civil-law countries. China is a typical civil-law country with great emphasis on the authority of law. Therefore, investor protection in China is relatively weak from the perspective of legal origin. Besides, the shareholder structure of China's listed companies presents a highly concentrated feature. However, the stock market of China is a retail market dominated by minority shareholders [10,16]. Minority shareholders have the characteristics of a weak ability to protect legal rights and an insufficient ability to exercise individual interests; they often face information asymmetry, and they are always victims of fraudulent issuance, internal transactions, false statements, and market manipulation by issuers [6]. The second type of agency problem between controlling shareholders and minority shareholders is more serious in China [9]. Therefore, China needs to set up institutions to protect the rights of minority shareholders.

In practice, the China Securities Investor Services Center (CSISC) was established in Shanghai in December 2014. It is a public welfare institution for securities and finance under the direct management of the China Securities Regulatory Commission (CSRC). Its businesses include exercise services, dispute mediation, rights protection services, investor education, investigation, and monitoring. In February 2016, the CSRC announced that the CSISC would carry out pilot programs in Shanghai, Guangdong (excluding Shenzhen), and Hunan. The main content of the pilot program was that CSISC holds 100 ordinary shares of listed companies and is not allowed to purchase more shares or trade any shares aiming to exercise rights as the minority shareholder, which gave CSISC the same legal status and rights as other shareholders [5,17]. CSISC can participate in corporate governance and exercise shareholder rights as minority shareholders, and it can also lead minority investors to actively exercise and safeguard rights according to laws like *Corporation Law* and *Security Law* through their exemplary roles to standardize the governance of listed companies. The rights protection services of the CSISC mainly include special representative litigation, support litigation, and shareholder litigation. Special representative litigation means that

CSISC is commissioned by more than 50 shareholders and represented in securities civil proceedings. Supporting litigation means that CSISC, as a supporting institution, selects cases, appoints litigation agents, and supports minority investors with impaired rights and interests in litigation according to law. Shareholder litigation refers to the litigation brought by shareholders to protect the legitimate rights and interests of the company or itself, including shareholder subrogation litigation, shareholder direct litigation, and so on. Investor education refers to public welfare publicity and education for investors, including publicity about "knowing rights, exercising rights, and safeguarding rights", basic knowledge of securities and futures, and the development of investor education products [18].

In essence, it is a new form of government public service institution and government supervision of listed companies. It has the characteristics of public interest shareholding by institutional investors and administrative characteristics, and it can bring positive externalities to society as public goods. After years of pilot and operation, the CSISC has become an important channel for the protection of small and medium investors and an important regulatory institution for the governance of listed companies in China [5,17,18].

The existing research exploring the impact of CSISC on the capital market is relatively limited. Hu et al. [18] pointed out that the pilot reform of CSISC shareholding can significantly reduce the stock price crash risks and promote the smooth operation of the financial market. It indicates that CSISC shareholding encourages minority shareholders' activism and plays a role in protecting the rights and interests of minority shareholders. Chen et al. [17] found that CSISC shareholding can reduce underinvestment but will not lead to overinvestment. Ge et al. [5] supposed that the CSISC shareholding system has awakened awareness of the rights protection of minority shareholders. Through the quasi-natural experiment of the CSISC shareholding pilot, they found that CSISC shareholding can strengthen corporate governance, reduce earnings management, and improve earnings quality. Based on the existing literature, this paper will further focus on how CSISC can improve the long-term sustainability performance of companies, which means the impact on ESG performance.

### 2.2. ESG

ESG is the abbreviation for Environment, Social Responsibility, and Governance. It is an investment philosophy and corporate rating standards with attention to environmental, social responsibility, and corporate governance performance rather than traditional financial performance. For enterprise ESG performance and the influencing factors of ESG performance, the existing research is mainly divided into internal and external factors of the company [19–21].

From the internal perspective of the company, the behaviors of corporate managers and controlling shareholders are the core factors in corporate strategy formulation. Battisti et al. [22] indicated that corporate venture capital can create a competitive advantage for enterprises and reflect it through the optimization of ESG performance. Barros et al. [23] found that corporate M&A activity has a positive effect on ESG scores. Research has also shown that there is a positive relationship between corporate non-financial information disclosure and corporate sustainability performance [24,25]. From a strategic perspective, the company's sustainable behaviors and projects [26], establishing a sustainability committee [27], customer and shareholder interest-oriented corporate strategies [28,29], and the digital development of a firm [30] lead to higher ESG scores and better ESG performance for companies.

From the development of external supervision, audit independence, and auditor experience [31], the presence of institutional investors [19,21,32] has a positive impact on firm ESG performance by improving the quality of accounting information, increasing investment in environmental protection, and increasing media attention, while audit committee tenure has a negative impact [33]. In terms of the countries and regions where the company is located, Mu et al. [34] indicated that the degree of digital finance development at the city

level where the company is located can increase the ESG score of the company by alleviating the financial constraints of the company. Besides, the country's competitiveness shock [35], government corruption [36], the development of the economy and culture [37,38], and green credit guidance policy [39] have a significant impact on the performance of ESG. Moreover, factors in the financial market such as investor sentiment are also the driving force behind corporate social responsibility [40].

Previous studies have shown that the behavior of minority shareholders plays an important role in the development of the long-term interests of enterprises. By obtaining investor comments in stock forum posts through the Java program to quantify minority shareholder activities (MSA), Xu et al. [10] showed that minority shareholders' activities have a positive effect on corporate social responsibility. From the perspective of investor protection policy, the online voting measures of listed companies in 2014 can enable minority shareholders to improve corporate social responsibility performance through internal control and increasing corporate transparency [9]. Although CSISC is a minority shareholder for companies, its policy and regulatory characteristics make its role different from that of ordinary minority shareholders, which is of special significance for corporate governance and investor protection [5,7,17]. So, our paper focuses on the relationship between the CSISC, as an institution of investor protection, and corporate social performance and ESG rating.

## 3. Hypothesis Development

Minority shareholders have insufficient ability to exercise individual interests, often face information asymmetry, and are always victims of fraudulent issuance, internal transactions, false statements, and market manipulation by issuers [6]. It is hard for minority shareholders to benefit from the short-term zero-sum game [8], which means minority shareholders can only obtain benefits through long-term value investment. Hence, it is vitally important for minority shareholders to protect their legal rights and gain benefits by focusing on ESG performance as a long-term sustainable development of companies [9,10].

The CSISC, as a minority shareholder, has a positive influence on ESG from two aspects. On one hand, CSISC shareholders could increase the supervision capability. The CSISC is an established organization affiliated with the CSRC [5], representing the attitude of the regulator as an innovative government regulation. Not only is the CSISC equipped with more competent professionals, but it is also able to cooperate deeply with regulators and the judicial system. Through judicial proceedings, the CSISC creates a deterrent effect on listed companies that attempt to violate the interests of minority shareholders through internal transactions and information fraud. Therefore, with strong external supervision, the CSISC can improve corporate governance and regulate the internal operating mechanisms of listed companies. Besides, the presence of the CSISC also arouses the awareness of minority shareholders to actively participate in the exercise of their rights and increases their chances of attending shareholders' meetings. As outsider investors, minority shareholders cannot accurately realize the financial information of listed companies. Instead, they focus on the long-term sustainable development ability of a company through the information disclosed on social responsibility, environmental protection, innovation investment, and other aspects. Thus, the negative news of social responsibility and environmental protection is more likely to attract the attention of minority shareholders. This strong supervision creates reputational pressure on companies, which encourages insiders to engage in valuable long-term strategic plans and focus on sustainable development, especially by reducing corporate socially irresponsible activities and avoiding penalties for environmental violations [31]. As ESG performance can reflect corporate long-term development and social and environmental responsibility by stimulating listed companies to pursue a more socially responsible approach and avoid misappropriation and opportunistic behavior to increase long-term value, we believe that the CSISC can prompt corporate ESG performance through its supervisory effect.

On the other hand, the CSISC would force firms to respond or disclose more information to avoid further investigation and penalties. The CSISC has filed some litigation cases involving information disclosure irregularities of listed companies, which shows that the quality of information disclosure of listed companies is a special concern of the CSISC at present. In the event of false statements, material omissions, and other illegal violations of listed companies, the CSISC can, through continuous tracking of the listed company's information disclosure, expose the problems of the listed company, timely release signals to the market, and increase the cost of information disclosure violations. Information disclosure is not only a way to alleviate information asymmetry and agency problems to increase corporate governance but also an important channel to actively display the development direction and social responsibility behavior of enterprises. When listed companies have a high level of information transparency and voluntary disclosure of non-financial information, they are more willing to actively disclose information related to the company's long-term development, including sustainability, corporate governance, and social and environmental responsibility, leading to a positive effect on ESG performance [24,41]. We believe that the CSISC can prompt corporate ESG performance by increasing information quality.

Based on the above discussion, we have developed the following hypothesis:

**Hypothesis 1 (H1):** *The pilot reform of CSISC shareholding has a positive influence on the ESG performance of listed companies.*

## 4. Research Design

### 4.1. Sample and Data

To construct the DID model to test and estimate the effect of the CSISC shareholding pilot reform in 2016, we selected a sample of listed firms registered in Shanghai, Zhejiang, Guangdong, Hunan, and Hubei provinces from 2013 to 2017, following the existing studies from Ge et al. [5] and Chen et al. [17]. According to these studies [42–44], ESG performance is measured by Huazheng ESG evaluation. Based on the core connotation and development experience of ESG, combined with the actual situation of the domestic market, the Huazheng ESG evaluation system constructs a three-level index system from top to bottom. Specifically, there are 3 first-level indicators, 14 second-level indicators, 26 third-level indicators, and more than 130 underlying data indicators. Through the combination of quarterly regular evaluation and dynamic tracking, the ESG score level of all A-share listed companies in the past 10 years was systematically measured (with a total score of 100 points), and nine ratings of "AAA-C" were given accordingly. Other data came from the CSMAR (China Stock Market and Accounting Research Database).

To ensure the comparability and similarity between samples, we further screened the samples as follows. First, because of the differences between financial companies and manufacturing companies in the statement structure, accounting treatment, audit requirements, and regulatory provisions, we omitted the financial companies. Second, pure B shares are subscribed and traded in foreign currencies, so we only used data from companies listed in the A-share market. Third, as the securities of ST (special treatment) companies cannot be circulated and traded normally in the market, nor can they be held by CSISC, so we omitted data on ST companies. Forth, we omitted samples with missing data. After selection and screening, we obtained 5264 firm-year observations.

### 4.2. Research Methodology

The research model is designed as follows:

$$ESG_{i,t+1} = \beta_0 + \beta_1 Treat_{i,t} \times Post_{i,t} + \beta_2 Treat_{i,t} + \beta_3 Post_{i,t} + Controls_{i,t} + Firm + Year + \varepsilon_{i,t} \tag{1}$$

The dependent variable $ESG_{i,t+1}$ indicates the corporate ESG score in year $t + 1$, which is the comprehensive and quantitative measure of corporate ESG performance. Considering Huazheng ESG rating is divided into nine levels from AAA to C to reflect the corporate ESG performance, we assigned this variable from C to AAA as 1–9 according to

the rating. As the policy may not have an immediate effect in that year, we used the ESG score of the next year to study the effects on corporate social responsibility performance. The independent variable $Treat_{i,t} \times Post_{i,t}$ reflects the effect of the pilot reform of CSISC shareholding. $Treat_{i,t}$ is a dummy variable indicating whether the firm is in the pilot region (Shanghai, Guangdong excluding Shenzhen and Hunan). Considering the economic and geographic factors to make the other conditions of the control group and the treated group close to the natural experiment, we chose listed companies in Zhejiang, Shenzhen, and Hubei as the control group. So, $Treat_{i,t}$ equals one for companies located in Shanghai, Guangdong, excluding Shenzhen and Hunan, and equals zero for companies located in Zhejiang, Shenzhen, and Hubei. The coefficient $\beta_1$ is our main interest. If a significant positive coefficient is observed, our hypothesis is confirmed, which indicates that CSISC shareholding promotes corporate ESG performance.

Following existing literature in the corporate finance [21,32,33,44,45], we controlled other variables influencing corporate social responsible behavior and ESG performance, including firm size (*Size*), asset-liability ratio (*Lev*), return on asset (*ROA*), price/book value ratio (*PB*), market performance (*TobinQ*), board size (*Boardsize*), board independence (*Indep*), CEO duality (*Duality*), ownership concentration (*Top10*), profitability (*Loss*), and firm age (*Age*). We also controlled firm-fixed effects and year-fixed effects in our model to reduce estimation bias. Appendix A provides a detailed definition of all the variables.

### 4.3. Descriptive Statistics

Table 1 shows the summary statistics of the main variables in the regression test. The mean of *Treat × Post* is 0.214, which means that 21.4% of observations were affected by the CSISC shareholding. On average, 39.6% of sample companies' assets are financed by liabilities; the market value is 2.066 times the book value. Approximately, each firm has a total number of seats on the Board of Directors on average, and 32.4% of the observations exist in the situation where the two positions of chairman and general manager are held by one person.

**Table 1.** Statistical description.

| Variable | Obs | Mean | Std | Min | P25 | Median | P75 | Max |
|---|---|---|---|---|---|---|---|---|
| *ESG* | 5264 | 6.480 | 1.067 | 4.000 | 6.000 | 6.000 | 7.000 | 9.000 |
| *Treat × Post* | 5264 | 0.214 | 0.410 | 0.000 | 0.000 | 0.000 | 0.000 | 1.000 |
| *Treat* | 5264 | 0.463 | 0.499 | 0.000 | 0.000 | 0.000 | 1.000 | 1.000 |
| *Post* | 5264 | 0.466 | 0.499 | 0.000 | 0.000 | 0.000 | 1.000 | 1.000 |
| *Size* | 5264 | 21.989 | 1.238 | 19.551 | 21.081 | 21.836 | 22.695 | 26.109 |
| *Lev* | 5264 | 0.396 | 0.198 | 0.046 | 0.234 | 0.382 | 0.539 | 0.897 |
| *ROA* | 5264 | 0.050 | 0.055 | −0.192 | 0.020 | 0.047 | 0.078 | 0.220 |
| *PB* | 5264 | 2.066 | 1.743 | 0.152 | 1.040 | 1.602 | 2.490 | 14.995 |
| *TobinQ* | 5264 | 2.460 | 1.678 | 0.885 | 1.472 | 1.956 | 2.800 | 15.212 |
| *Boardsize* | 5264 | 8.475 | 1.645 | 5.000 | 7.000 | 9.000 | 9.000 | 15.000 |
| *Indep* | 5264 | 0.376 | 0.054 | 0.333 | 0.333 | 0.357 | 0.429 | 0.571 |
| *Duality* | 5264 | 0.324 | 0.468 | 0.000 | 0.000 | 0.000 | 1.000 | 1.000 |
| *Top10* | 5264 | 0.161 | 0.110 | 0.015 | 0.077 | 0.135 | 0.219 | 0.575 |
| *Loss* | 5264 | 0.063 | 0.242 | 0.000 | 0.000 | 0.000 | 0.000 | 1.000 |
| *Age* | 5264 | 2.058 | 0.850 | 0.048 | 1.518 | 2.082 | 2.864 | 3.263 |

*Notes*: This table reports the descriptive statistics of the main variables for samples from 2013–2017. See Appendix A for the definition of all variables.

Table 2 reports the Pearson correlation matrix for the variables used in the regression test. The independent variable *Treat × Post* is significantly and positively correlated with the ESG score, and the result is the preliminary validation of the hypothesis, but more accurate estimation and inference still need further empirical tests.

**Table 2.** Pearson correlation matrix.

| | (A) | (B) | (C) | (D) | (E) | (F) | (G) | (H) | (I) | (J) | (K) | (L) | (M) | (N) | (O) |
|---|---|---|---|---|---|---|---|---|---|---|---|---|---|---|---|
| (A)ESG | 1.000 | | | | | | | | | | | | | | |
| (B)Treat × Post | 0.023 * | 1.000 | | | | | | | | | | | | | |
| (C)Treat | 0.063 *** | 0.561 *** | 1.000 | | | | | | | | | | | | |
| (D)Post | −0.035 ** | 0.558 *** | −0.009 | 1.000 | | | | | | | | | | | |
| (E)Size | 0.418 *** | 0.052 *** | 0.044 *** | 0.062 *** | 1.000 | | | | | | | | | | |
| (F)Lev | 0.158 *** | −0.030 ** | −0.007 | −0.049 *** | 0.539 *** | 1.000 | | | | | | | | | |
| (G)ROA | 0.171 *** | 0.033 ** | −0.034 ** | 0.105 *** | −0.010 | −0.325 *** | 1.000 | | | | | | | | |
| (H)PB | −0.180 *** | −0.044 *** | −0.005 | −0.072 *** | −0.481 *** | −0.337 *** | 0.058 *** | 1.000 | | | | | | | |
| (I)TobinQ | −0.167 *** | −0.048 *** | −0.005 | −0.080 *** | −0.434 *** | −0.234 *** | 0.023* | 0.994 *** | 1.000 | | | | | | |
| (J)Boardsize | 0.177 *** | −0.047 *** | 0.005 | −0.067 *** | 0.302 *** | 0.197 *** | −0.042 *** | −0.173 *** | −0.157 *** | 1.000 | | | | | |
| (K)Indep | −0.029 ** | 0.048 *** | 0.047 *** | 0.018 | −0.033 ** | −0.036 *** | −0.010 | 0.067 *** | 0.066 *** | −0.514 *** | 1.000 | | | | |
| (L)Duality | −0.118 *** | −0.017 | −0.073 *** | 0.054 *** | −0.192 *** | −0.146 *** | 0.069 *** | 0.066 *** | 0.051 *** | −0.186 *** | 0.126 *** | 1.000 | | | |
| (M)Top10 | 0.122 *** | −0.018 | 0.013 | −0.040 *** | 0.189 *** | 0.023 * | 0.173 *** | −0.134 *** | −0.135 *** | 0.000 | 0.053 *** | 0.002 | 1.000 | | |
| (N)Loss | −0.143 *** | −0.031 ** | 0.009 | −0.064 *** | −0.049 *** | 0.143 *** | −0.518 *** | 0.062 *** | 0.080 *** | 0.018 | 0.027 * | −0.031 ** | −0.070 *** | 1.000 | |
| (O)Age | 0.204 *** | −0.011 | 0.066 *** | −0.078 *** | 0.442 *** | 0.398 *** | −0.253 *** | −0.001 | 0.045 *** | 0.199 *** | −0.068 *** | −0.249 *** | −0.151 *** | 0.113 *** | 1.000 |

*Notes:* This table reports the Pearson correlation matrix for the variables used in our test, where *, **, and *** denote significance levels of 0.10, 0.05, and 0.01, respectively.

## 5. Empirical Results

### 5.1. Baseline Results

To test our hypothesis, we estimated a baseline regression model with Equation (1). Table 3 reports the baseline results with column (1) without control variables and column (2) with all eleven control variables. As the results show, the coefficient of *Treat × Post* with all (no) control variables is 0.091 (0.086) and significant at a 1% (5%) level, suggesting that after the CSISC shareholding pilot reform, the ESG score in pilot regions increased by 0.091 on average. The results are also significant economically. When it comes to the control variables, the control variables *ROA* and *Duality* are positively and significantly related to the ESG score, and variables such as *Loss* and *Age* have a significantly negative relationship with the ESG score. Our study also shows that firms with higher profitability, shorter listed times, and a corporate governance strategy with core position duality have higher ESG scores. Overall, the results in Table 3 indicate that the CSISC can prompt corporate ESG performance through increased supervisory effect and information quality, which is consistent with our hypothesis.

**Table 3.** Baseline results—regulatory minority shareholder and ESG performance.

| | (1) *ESG* | (2) *ESG* |
|---|---|---|
| *Treat × Post* | **0.086 \*\*** | **0.091 \*\*\*** |
| | **(2.47)** | **(2.60)** |
| *Treat* | 0.441 | 0.599 |
| | (1.06) | (1.44) |
| *Post* | −0.016 | −0.013 |
| | (−0.52) | (−0.33) |
| *Size* | | 0.048 |
| | | (1.48) |
| *Lev* | | 0.208 |
| | | (0.68) |
| *ROA* | | 1.184 \*\*\* |
| | | (3.84) |
| *PB* | | 0.184 |
| | | (0.62) |
| *TobinQ* | | −0.210 |
| | | (−0.71) |
| *Boardsize* | | −0.007 |
| | | (−0.49) |
| *Indep* | | −0.321 |
| | | (−0.90) |
| *Duality* | | 0.086 \*\* |
| | | (2.40) |
| *Top10* | | 0.134 |
| | | (0.48) |
| *Loss* | | −0.083 \* |
| | | (−1.69) |
| *Age* | | −0.225 \*\*\* |
| | | (−3.14) |
| *Constant* | 6.241 \*\*\* | 5.735 \*\*\* |
| | (32.07) | (7.71) |
| *Year & Firm* | Yes | Yes |
| *Observations* | 5264 | 5264 |
| *Adjusted R²* | 0.004 | 0.023 |

*Notes:* This table reports the baseline results of this study, with the first column without firm-level controls and the second column with all the controls. The dependent variable is *ESG*. The independent variable is *Treat × Post*. Firm-fixed effects and year-fixed effects are also included. The *t* values reported in parentheses are adjusted based on robust standard errors, where *, **, and *** denote significance levels of 0. 1, 0.05, and 0.01, respectively. All variables are defined in Appendix A.

*5.2. Robustness Tests*

5.2.1. Parallel Trends Analysis

A parallel trend is a basic assumption for the difference-in-differences method, which means that the treated group and control group follow a parallel trend before the exogenous shock. Otherwise, the difference between the treated group and the control group after the policy and coefficient of *Treat × Post* not only include the policy impact but also the difference between the treated group and the control group. To validate the parallel trend, we introduced three dummies into the baseline regression:

$$ESG_{i,t+1} = \alpha_0 + \alpha_1 Pre\_1_{i,t} + \alpha_2 Current_{i,t} + \alpha_3 Post\_1_{i,t} + Controls_{i,t} + Firm + Year + u_{i,t} \quad (2)$$

The dependent variable is still the ESG score of the company i in year t + 1. *Pre_1* represents the multiplication of the dummy where the sample is in 2015 (one year before CSISC shareholding pilot reform) and *Treat*; *Current* represents the multiplication of the dummy where the sample is in 2016 (the starting year of CSISC shareholding pilot reform) and *Treat*; *Post_1* represents the multiplication of the dummy where the sample is in 2017 (one year after CSISC shareholding pilot reform) and *Treat*. Firm and year-fixed effects are included in the regression model. Table 4 reports the results of the parallel trend analysis. The coefficient of *Pre_1* is not statistically significant, whereas the coefficients of *Current* and *Post_1* are positive and significant. Moreover, the value of the coefficient of *Post_1* is larger than that of *Current*, which indicates that the CSISC shareholding pilot policy's effect on corporate ESG performance is continuous and intensified.

**Table 4.** Parallel trend.

|  | (1) *ESG* |
| --- | --- |
| *Pre_1* | 0.068 |
|  | (1.45) |
| *Current* | 0.113 ** |
|  | (2.44) |
| *Post_1* | 0.125 *** |
|  | (2.68) |
| *Size* | 0.050 |
|  | (1.53) |
| *Lev* | 0.198 |
|  | (0.65) |
| *ROA* | 1.187 *** |
|  | (3.84) |
| *PB* | 0.180 |
|  | (0.60) |
| *TobinQ* | −0.206 |
|  | (−0.69) |
| *Boardsize* | −0.007 |
|  | (−0.48) |
| *Indep* | −0.333 |
|  | (−0.93) |
| *Duality* | 0.085 ** |
|  | (2.39) |
| *Top10* | 0.118 |
|  | (0.42) |
| *Loss* | −0.080 |
|  | (−1.63) |
| *Age* | −0.220 *** |
|  | (−3.06) |
| *Constant* | 5.944 *** |
|  | (8.11) |
| *Year & Firm* | Yes |
| *Observations* | 5264 |
| *Adjusted R²* | 0.023 |

*Notes:* This table reports the results of the parallel trends, with all the controls. The dependent variable is *ESG*. The independent variable is *Pre_1, Current, and Post_1*. Firm-fixed effects and year-fixed effects are also included. The *t* values reported in parentheses are adjusted based on robust standard errors, where *, **, and *** denote significance levels of 0.10, 0.05, and 0.01, respectively. All variables are defined in Appendix A.

### 5.2.2. Placebo Test

To reduce the endogenous problems of the study and enhance the robustness of the results, following Huang et al. [20] and Song et al. [46], we conducted placebo tests by changing the policy's time to an earlier year, 2012, and the sample time period to 2009–2013. The variable *FPost* equals one if the observation is in 2012 and 2013, and equals zero if the observation is in 2009–2011. Variable *Treat × FPost* is the multiplication of *Treat* and *FPost*. The results of the placebo test are shown in Table 5; the coefficient of *Treat × FPost*, *Treat*, and *FPost* are not significant, at least at the 10% level. The counterfactual placebo test confirms the main hypothesis.

**Table 5.** Placebo test.

|  | **(1) *ESG*** |
|---|---|
| ***Treat × FPost*** | **0.027** |
|  | **(0.67)** |
| *Treat* | 0.222 |
|  | (0.48) |
| *FPost* | 0.021 |
|  | (0.47) |
| *Size* | 0.234 *** |
|  | (4.19) |
| *Lev* | −0.437 |
|  | (−0.98) |
| *ROA* | 0.143 |
|  | (0.39) |
| *PB* | 0.358 |
|  | (0.84) |
| *TobinQ* | −0.338 |
|  | (−0.79) |
| *Boardsize* | 0.010 |
|  | (0.54) |
| *Indep* | −0.556 |
|  | (−1.21) |
| *Duality* | 0.022 |
|  | (0.47) |
| *Top10* | −0.311 |
|  | (−0.78) |
| *Loss* | −0.077 |
|  | (−1.28) |
| *Age* | −0.111 * |
|  | (−1.80) |
| *Constant* | 2.027 |
|  | (1.59) |
| *Year & Firm* | Yes |
| *Observations* | 3724 |
| *Adjusted $R^2$* | 0.045 |

*Notes:* This table reports the results of the placebo test, with all the controls. The dependent variable is *ESG*. The independent variable is *Treat × FPost*. Firm-fixed effects and year-fixed effects are also included. The *t* values reported in parentheses are adjusted based on robust standard errors, where *, **, and *** denote significance levels of 0.10, 0.05, and 0.01, respectively. All variables are defined in Appendix A.

### 5.2.3. Propensity Score Matching (PSM)

To alleviate the endogeneity from potential self-selection bias, we conducted a propensity score matching (PSM) before regression. The advantage of PSM-DID is that we can compare firms with the most similar characteristics. To be specific, we used an unrepeatable sampling of 1:1 nearest-neighbor matching with control variables in the baseline regression as covariates. Then, we used the 4508 observations after matching into the regression of Equation (1). The results are shown in Table 6; the coefficient of *Treat × Post* with all control variables was 0.102 and significant at a 1% level, suggesting that CSISC shareholding pilot

reform has a positive effect on improving corporate ESG performance. Moreover, the coefficient of PSM-DID was 0.102, which has no big gap with the estimated coefficient in Table 3 (0.091). So, our hypothesis still holds for PSM samples.

**Table 6.** PSM-DID.

|  | (1) *ESG* |
|---|---|
| *Treat* × *Post* | 0.102 *** |
|  | (2.68) |
| *Treat* | −0.020 |
|  | (−0.03) |
| *Post* | −0.032 |
|  | (−0.74) |
| *Size* | 0.039 |
|  | (1.09) |
| *Lev* | 0.206 |
|  | (0.55) |
| *ROA* | 1.005 *** |
|  | (3.04) |
| *PB* | 0.287 |
|  | (0.78) |
| *TobinQ* | −0.318 |
|  | (−0.87) |
| *Boardsize* | 0.001 |
|  | (0.08) |
| *Indep* | 0.191 |
|  | (0.47) |
| *Duality* | 0.095 ** |
|  | (2.37) |
| *Top10* | 0.515 * |
|  | (1.66) |
| *Loss* | −0.090 * |
|  | (−1.69) |
| *Age* | −0.199 ** |
|  | (−2.50) |
| *Constant* | 5.911 *** |
|  | (6.85) |
| *Year & Firm* | Yes |
| *Observations* | 4508 |
| *Adjusted $R^2$* | 0.024 |

Notes: This table reports the results of PSM-DID, with all the controls. The dependent variable is *ESG*. The independent variable is *Treat* × *Post*. Firm-fixed effects and year-fixed effects are also included. The *t* values reported in parentheses are adjusted based on robust standard errors, where *, **, and *** denote significance levels of 0.10, 0.05, and 0.01, respectively. All variables are defined in Appendix A.

### 5.2.4. Alternative Control Group

Taking geographical and economic factors into account, we used data from firms registered in Shanghai, Zhejiang, Guangdong, Hunan, and Hubei provinces as observations in the baseline regression. In this section, we change the definition of the control group and observations as a robustness test. On the basis of the study of Ge et al. [5], we chose Jiangsu and Zhejiang as control regions for Shanghai, Hubei as the control province for Hunan, and Shenzhen as the control region for Guangdong (excluding Shenzhen). We redefined the variable *Treat2*; if the observation is in Shanghai, Hunan, and Guangdong (excluding Shenzhen), *Treat2* equals one; if the observation is in Jiangsu, Zhejiang, Hubei, and Shenzhen, *Treat2* equals zero. The results are shown in Table 7; the coefficient of *Treat2* × *Post* is was significantly positive ($\beta = 0.074$, $p < 0.05$). So, our hypothesis still holds after changing the control group.

**Table 7.** Alternative control group.

|  | **(1) ESG** |
| --- | --- |
| ***Treat2 × Post*** | **0.074 **** |
|  | **(2.34)** |
| *Treat2* | 0.163 |
|  | (0.47) |
| *Post* | −0.007 |
|  | (−0.21) |
| *Size* | 0.053 * |
|  | (1.84) |
| *Lev* | 0.171 |
|  | (0.60) |
| *ROA* | 1.290 *** |
|  | (4.64) |
| *PB* | 0.156 |
|  | (0.56) |
| *TobinQ* | −0.175 |
|  | (−0.62) |
| *Boardsize* | −0.014 |
|  | (−0.98) |
| *Indep* | −0.478 |
|  | (−1.47) |
| *Duality* | 0.065 ** |
|  | (2.06) |
| *Top10* | 0.426 * |
|  | (1.82) |
| *Loss* | −0.074 * |
|  | (−1.69) |
| *Age* | −0.173 *** |
|  | (−2.76) |
| *Constant* | 5.766 *** |
|  | (8.91) |
| *Year & Firm* | Yes |
| *Observations* | 6585 |
| *Adjusted R²* | 0.021 |

*Notes:* This table reports the results with an alternative control group, with all the controls. The dependent variable is *ESG*. The independent variable is *Treat2 × Post*. Firm-fixed effects and year-fixed effects are also included. The *t* values reported in parentheses are adjusted based on robust standard errors, where *, **, and *** denote significance levels of 0.10, 0.05, and 0.01, respectively. All variables are defined in Appendix A.

### 5.2.5. Other Robustness Tests

To control the effects of other exogenous shocks, industry differences, and regional effects, we conducted a series of robustness tests as follows. First, because China's capital market experienced a serious stock disaster in 2015, the market performance and sample data in 2015 are abnormal. Hence, we omitted samples from the year 2015; the results are shown in column (1) in Table 8, and the coefficient of *Treat×Post* is still significantly positive ($\beta = 0.106$, $p < 0.05$). Second, the CSISC shareholding was extended to the whole country in May 2017; hence, to improve the accuracy of estimation, we omitted observations in 2017 and redefined the meaning of the variable *Post*. If the observation is in the year 2016, *Post* equals one, otherwise, *Post* equals zero. The results from the redefined *Post* are shown in column (2) in Table 8, and the coefficient of *Treat × Post* is still significantly positive ($\beta = 0.079$, $p < 0.10$). Last, to control the influence of unknown factors that change with industries and provinces, we added the cross-fixed effect of industries and provinces in the regression. The results from the redefined post are shown in column (3) in Table 8, and the coefficient of *Treat × Post* is still significantly positive ($\beta = 0.087$, $p < 0.05$). In short, the results still hold true after considering time and regional effects.

**Table 8.** Other robustness tests.

| | **(1)** *ESG* | **(2)** *ESG* | **(3)** *ESG* |
|---|---|---|---|
| *Treat × Post* | **0.106 **** | **0.079 *** | **0.087 **** |
| | **(2.57)** | **(1.91)** | **(2.48)** |
| *Treat* | 0.610 | 1.081 * | −0.190 |
| | (1.32) | (1.68) | (−0.29) |
| *Post* | −0.048 | 0.027 | −0.015 |
| | (−0.98) | (0.76) | (−0.38) |
| *Size* | 0.045 | 0.033 | 0.059 * |
| | (1.21) | (0.85) | (1.73) |
| *Lev* | 0.120 | 0.365 | 0.126 |
| | (0.35) | (1.07) | (0.40) |
| *ROA* | 1.492 *** | 0.847 ** | 1.126 *** |
| | (4.07) | (2.36) | (3.63) |
| *PB* | 0.039 | 0.291 | 0.241 |
| | (0.11) | (0.87) | (0.78) |
| *TobinQ* | −0.085 | −0.318 | −0.269 |
| | (−0.25) | (−0.94) | (−0.87) |
| *Boardsize* | −0.011 | 0.005 | −0.011 |
| | (−0.62) | (0.31) | (−0.71) |
| *Indep* | −0.378 | −0.099 | −0.323 |
| | (−0.89) | (−0.24) | (−0.90) |
| *Duality* | 0.100 ** | 0.073 * | 0.067 * |
| | (2.41) | (1.76) | (1.87) |
| *Top10* | 0.133 | −0.038 | 0.276 |
| | (0.42) | (−0.12) | (0.97) |
| *Loss* | −0.078 | −0.054 | −0.092 * |
| | (−1.29) | (−1.02) | (−1.89) |
| *Age* | −0.189 ** | −0.380 *** | −0.199 *** |
| | (−2.25) | (−3.91) | (−2.77) |
| *Constant* | 5.773 *** | 6.059 *** | 11.090 *** |
| | (6.84) | (6.63) | (6.83) |
| *Year & Firm* | Yes | Yes | Yes |
| *Industry & Province* | No | No | Yes |
| *Observations* | 4252 | 3917 | 5264 |
| *Adjusted R²* | 0.026 | 0.023 | 0.047 |

*Notes:* This table reports the results of time and regional effects, with all the controls. The dependent variable is *ESG*. Firm-fixed effects and year-fixed effects are also included. The independent variable is *Treat × Post*. Column (1) is the results without observations in 2015, column (2) is the results without observations in 2017, and column (3) is the results with a fixed effect of industry and province. The *t* values reported in parentheses are adjusted based on robust standard errors, where *, **, and *** denote significance levels of 0.10, 0.05, and 0.01, respectively. All variables are defined in Appendix A.

### 5.3. Heterogeneity Analysis

5.3.1. Size Heterogeneity

The size of an enterprise will affect its development mode, investing, and financing capacity as well as strategies for responding to policy changes. The larger firms have more resources and more often use reporting tools to provide ESG data [47].

So, the influence of CSISC shareholding may be different between companies of different sizes. The samples are divided into two groups according to the median of the total asset logarithms of enterprises. Those above the median are large enterprises, while those above the median are small enterprises. Column (1) and (2) in Table 9 reveal the result, column (1) is the result for large enterprises and column (2) is the result for small enterprises. For column (1), the coefficient of *Treat × Post* is significantly positive, which means CSISC shareholding has a positive influence on large enterprises statistically. Conversely, for small enterprises, to coefficient is not significant, so, the policy has no impact on small enterprises statistically.

**Table 9.** Heterogeneity analysis—company size and industry type.

| | (1) ESG Large-Sized | (2) ESG Small-Sized | (3) ESG High-Polluting | (4) ESG Non-High-Polluting |
|---|---|---|---|---|
| **Treat × Post** | **0.105 **** | **0.057** | **0.124** | **0.084 **** |
| | **(2.12)** | **(1.03)** | **(1.64)** | **(2.13)** |
| Treat | 1.152 ** | −0.294 | −0.174 | 1.139 ** |
| | (2.18) | (−0.47) | (−0.23) | (2.24) |
| Post | 0.001 | −0.088 | −0.070 | 0.011 |
| | (0.01) | (−1.34) | (−0.82) | (0.25) |
| Size | 0.151 ** | −0.028 | 0.105 | 0.030 |
| | (2.51) | (−0.42) | (1.32) | (0.83) |
| Lev | −0.342 | 0.057 | −0.038 | 0.357 |
| | (−0.43) | (0.16) | (−0.05) | (1.00) |
| ROA | 0.244 | 1.339 *** | 0.520 | 1.508 *** |
| | (0.50) | (3.07) | (0.85) | (4.18) |
| PB | −0.183 | 0.023 | 0.201 | 0.253 |
| | (−0.23) | (0.07) | (0.28) | (0.73) |
| TobinQ | 0.180 | −0.041 | −0.220 | −0.282 |
| | (0.23) | (−0.12) | (−0.31) | (−0.81) |
| Boardsize | −0.045 ** | 0.040 | 0.016 | −0.015 |
| | (−2.29) | (1.50) | (0.51) | (−0.87) |
| Indep | −1.108 ** | 0.437 | 0.060 | −0.493 |
| | (−2.14) | (0.80) | (0.09) | (−1.17) |
| Duality | 0.027 | 0.085 * | 0.208 *** | 0.056 |
| | (0.49) | (1.66) | (2.68) | (1.40) |
| Top10 | 0.568 | 0.201 | 0.682 | −0.056 |
| | (1.40) | (0.37) | (1.20) | (−0.17) |
| Loss | −0.125 * | −0.085 | −0.134 | −0.065 |
| | (−1.68) | (−1.30) | (−1.32) | (−1.17) |
| Age | −0.352 ** | −0.014 | −0.069 | −0.288 *** |
| | (−2.53) | (−0.15) | (−0.44) | (−3.62) |
| Constant | 4.418 *** | 6.349 *** | 4.087 ** | 6.197 *** |
| | (3.14) | (4.32) | (2.30) | (7.50) |
| Year & Firm | Yes | Yes | Yes | Yes |
| Observations | 2632 | 2632 | 1482 | 3782 |
| Adjusted $R^2$ | 0.024 | 0.041 | 0.024 | 0.029 |

*Notes:* This table reports the results of heterogeneity analysis, with all the controls. The dependent variable is *ESG*. The independent variable is *Treat × Post*. Firm-fixed effects and year-fixed effects are also included. Column (1) is the results of large-sized companies, column (2) is the results of small-sized companies, column (3) is the results of high-polluting companies, and column (4) is the results of non-high-polluting companies. The *t* values reported in parentheses are adjusted based on robust standard errors, where *, **, and *** denote significance levels of 0.10, 0.05, and 0.01, respectively. All variables are defined in Appendix A.

### 5.3.2. Polluting Heterogeneity

Different industries will have different ESG performances due to their differences in scale, development situation, and factor endowment. The polluting industry is not only the core part of pollution prevention and control but also for the realization of carbon peaking and carbon neutrality. Since the meaning of "E" in ESG is environmental, there may be differences in the effects of CSISC shareholding on ESG between high-polluting industries and non-high-polluting industries. The characteristics and definition of high-polluting industries are based on the following documents: *Notice on Environmental Protection Verification of Enterprises Applying for Listing* and *Listed Enterprises Applying for Refinancing and Notice on Further Regulating Environmental Protection Verification of Production and Operation Companies in Heavy Pollution Industries Applying for Listing or Refinancing*. The results are shown in Table 9, where column (3) is the regression for high-polluting samples and column (4) is the regression for non-high-polluting samples. The coefficient of high-polluting industries is not significant but significantly positive for non-high-polluting industries. So, the policy

has a positive effect on corporate ESG performance in non-high-polluting industries and no significant effect on that in high-polluting industries, statistically.

### 5.4. Cross-Sectional Analysis

5.4.1. Analyst Coverage

Previous studies showed that analyst coverage can improve CSR by increasing site visits from institutional investors and improving the firm's internal controls [18]. In this section, we explore whether analysts' attention influences the relationship between CSISC shareholding and corporate ESG performance. We used the number of research reports to quantify analysts' attention; if the number of research reports from the company exceeds the industry median, the variable *Datt* equals one, otherwise, it equals zero. Column (1) in Table 10 reveals the moderating effect of the analysts' attention. The coefficient of *Treat × Post × Datt* is significantly positive, which has the same sign as the coefficient of *Treat × Post* in baseline regression. So, analysts' attention strengthens the effect of CSISC shareholding on corporate ESG performance.

**Table 10.** Cross-sectional analysis—external governance environment.

|  | (1) *ESG* | (2) *ESG* | (3) *ESG* | (4) *ESG* |
|---|---|---|---|---|
|  | Analyst Coverage | External Auditor | Institutional Shareholders | Market Intermediary |
| *Treat × Post × Datt* | 0.143 * | | | |
|  | (1.90) | | | |
| *Treat × Datt* | 0.050 | | | |
|  | (0.81) | | | |
| *Post × Datt* | 0.032 | | | |
|  | (0.63) | | | |
| *Datt* | −0.025 | | | |
|  | (−0.59) | | | |
| *Treat × Post × Big4* | | 0.310 ** | | |
|  | | (1.98) | | |
| *Treat × Big4* | | 0.297 | | |
|  | | (1.12) | | |
| *Post × Big4* | | 0.057 | | |
|  | | (0.47) | | |
| *big4* | | −0.071 | | |
|  | | (−0.37) | | |
| *Treat × Post × DRIns* | | | 0.150 ** | |
|  | | | (2.05) | |
| *Treat × DRIns* | | | −0.004 | |
|  | | | (−0.05) | |
| *Post × DRIns* | | | 0.037 | |
|  | | | (0.74) | |
| *DRIns* | | | −0.010 | |
|  | | | (−0.18) | |
| *Treat × Post × Index* | | | | 0.034 ** |
|  | | | | (2.02) |
| *Treat × Index* | | | | 0.008 |
|  | | | | (0.42) |
| *Post × Index* | | | | −0.018 |
|  | | | | (−1.51) |
| *Index* | | | | −0.006 |
|  | | | | (−0.24) |
| *Treat × Post* | 0.015 | 0.068 * | 0.015 | −0.307 |
|  | (0.28) | (1.88) | (0.29) | (−1.55) |
| *Treat* | 0.595 | 0.591 | 0.606 | 0.496 |
|  | (1.43) | (1.42) | (1.45) | (1.07) |

**Table 10.** *Cont.*

|  | **(1) *ESG*** | **(2) *ESG*** | **(3) *ESG*** | **(4) *ESG*** |
|---|---|---|---|---|
|  | **Analyst Coverage** | **External Auditor** | **Institutional Shareholders** | **Market Intermediary** |
| *Post* | −0.017 | −0.025 | −0.039 | 0.184 |
|  | (−0.35) | (−0.62) | (−0.81) | (1.27) |
| *Size* | 0.027 | 0.056 * | 0.047 | 0.047 |
|  | (0.80) | (1.73) | (1.43) | (1.46) |
| *Lev* | 0.161 | 0.199 | 0.206 | 0.225 |
|  | (0.53) | (0.66) | (0.68) | (0.74) |
| *ROA* | 1.142 *** | 1.207 *** | 1.132 *** | 1.196 *** |
|  | (3.67) | (3.92) | (3.67) | (3.87) |
| *PB* | 0.123 | 0.189 | 0.183 | 0.206 |
|  | (0.41) | (0.64) | (0.62) | (0.69) |
| *TobinQ* | −0.150 | −0.216 | −0.210 | −0.233 |
|  | (−0.50) | (−0.73) | (−0.71) | (−0.78) |
| *Boardsize* | −0.009 | −0.004 | −0.005 | −0.007 |
|  | (−0.58) | (−0.26) | (−0.34) | (−0.48) |
| *Indep* | −0.352 | −0.288 | −0.341 | −0.347 |
|  | (−0.98) | (−0.81) | (−0.95) | (−0.97) |
| *Duality* | 0.086 ** | 0.086 ** | 0.082 ** | 0.085 ** |
|  | (2.42) | (2.41) | (2.30) | (2.37) |
| *Top10* | 0.163 | 0.111 | 0.120 | 0.126 |
|  | (0.58) | (0.39) | (0.41) | (0.45) |
| *Loss* | −0.090 * | −0.080 | −0.085* | −0.081 * |
|  | (−1.84) | (−1.63) | (−1.74) | (−1.66) |
| *Age* | −0.245 *** | −0.207 *** | −0.202 *** | −0.207 *** |
|  | (−3.39) | (−2.87) | (−2.81) | (−2.86) |
| *Constant* | 6.267 *** | 5.484 *** | 5.722 *** | 5.809 *** |
|  | (8.24) | (7.34) | (7.65) | (7.05) |
| *Year & Firm* | Yes | Yes | Yes | Yes |
| *Observations* | 5264 | 5264 | 5264 | 5264 |
| *Adjusted R²* | 0.027 | 0.027 | 0.026 | 0.024 |

*Notes:* This table reports the results of the cross-sectional analysis of this study, with all the controls. The dependent variable is *ESG*. Firm-fixed effects and year-fixed effects are also included. Column (1) shows the moderating effect of analyst coverage, column (2) shows the moderating effect of the external auditor, column (3) shows the moderating effect of institutional shareholder, and column (4) shows the effect of the market intermediary. The *t* values reported in parentheses are adjusted based on robust standard errors, where *, **, and *** denote significance levels of 0.10, 0.05, and 0.01, respectively. All variables are defined in Appendix A.

### 5.4.2. External Auditing

Auditing quality, as a prominent part of external governance, may have an impact on corporate ESG performance [31]. We used the variable *big4* as an indicator of whether the firm's annual report is prepared by the "big four" accounting firms. *Big4* is a dummy variable that equals one the annual report is prepared by the "big four" and zero otherwise. The results are shown in column (2) of Table 10. The coefficient of *Treat* × *Post* × *big4* is significantly positive, which has the same sign as the coefficient of *Treat* × *Post* in baseline regression. So, external auditing quality strengthens the effect of CSISC shareholding on corporate ESG performance.

### 5.4.3. Institutional Shareholders

An institutional investor may drive long-term incentives and improve information quality, thus affecting the ESG score [21,44]. To analyze the role of institutional shareholders, we define the moderating variable *DRIns*, which is equal to one if the shareholding ratio of the company's institutional investors is greater than the sample median and zero otherwise. The results are shown in column (3) of Table 10. The coefficient of *Treat* × *Post* × *DRIns* is significantly positive, which has the same sign as the coefficient of *Treat* × *Post* in baseline

regression. So, institutional shareholding strengthens the effect of CSISC shareholding on corporate ESG performance.

### 5.4.4. Market Intermediary

The development of market intermediary organizations (such as law firms and accountancy firms) and the improvement of legal systems are important elements of the capital market [48,49]. We used the variable index, which is the *Market Intermediary Index*, to define the development of the market intermediary and legal system. The results are shown in column (4) of Table 10. The coefficient of *Treat × Post × Index* is significantly positive, which has the same sign as the coefficient of *Treat × Post* in baseline regression. So, a high-developed market intermediary and legal system strengthen the effect of CSISC shareholding on corporate ESG performance.

Combining the results of the above regression, they support our argument that analyst coverage, external auditing, institutional shareholders, and the market intermediary will strengthen the positive relationship between CSISC shareholding and corporate ESG performance, consistent with our logical analysis.

### 5.5. Mechanisms Analysis

Based on the analysis in our hypothesis, we suppose that the CSISC, as a minority shareholder, leads a positive influence on the ESG through increasing supervision capability and information transparency. To test our logical derivation, we conducted a mechanism analysis.

We use the number of dissenting votes at the Annual Board of Directors to indicate the supervision capability. The results are shown in column (1) of Table 11; the coefficient of *Treat × Post* is significantly positive, which means that the CSISC shareholding has a positive impact on supervision capability and can improve the efficiency of the Board of Directors.

**Table 11.** Mechanisms analysis– supervision capability and information transparency.

|  | (1) *Supervision Capability* | (2) *Information Transparency* |
| --- | --- | --- |
| *Treat × Post* | 0.286 ** | −0.020 ** |
|  | (2.00) | (−2.06) |
| *Treat* | −0.354 | −0.020 |
|  | (−0.21) | (−0.15) |
| *Post* | 0.212 | −0.016 |
|  | (1.29) | (−1.41) |
| *Size* | 0.119 | 0.026 *** |
|  | (0.89) | (2.96) |
| *Lev* | 1.128 | 0.126 |
|  | (0.91) | (1.38) |
| *ROA* | −3.429 *** | 0.148 * |
|  | (−2.72) | (1.76) |
| *PB* | 2.748 ** | 0.073 |
|  | (2.27) | (0.83) |
| *TobinQ* | −2.607 ** | −0.070 |
|  | (−2.14) | (−0.79) |
| *Boardsize* | 0.119 * | −0.004 |
|  | (1.94) | (−0.96) |
| *Indep* | 4.955 *** | −0.090 |
|  | (3.39) | (−0.94) |
| *Duality* | −0.064 | 0.015 |
|  | (−0.44) | (1.52) |
| *Top10* | 0.077 | 0.039 |
|  | (0.07) | (0.52) |
| *Loss* | 0.361 * | 0.019 |
|  | (1.80) | (1.49) |

**Table 11.** *Cont.*

|  | **(1)** *Supervision Capability* | **(2)** *Information Transparency* |
|---|---|---|
| *Age* | −0.430 | 0.003 |
|  | (−1.47) | (0.12) |
| *Constant* | −3.826 | −0.479 ** |
|  | (−1.26) | (−2.32) |
| *Year & Firm* | Yes | Yes |
| *Observations* | 5264 | 4640 |
| *Adjusted $R^2$* | 0.017 | 0.050 |

*Notes:* This table reports the results of the mechanisms analysis, with all the controls. In column (1), the dependent variable is *Vote*, and in column (2), the dependent variable is *DA*. The independent variable is *Treat × Post*. Firm-fixed effects and year-fixed effects are also included. The *t* values reported in parentheses are adjusted based on robust standard errors, where *, **, and *** denote significance levels of 0.10, 0.05, and 0.01, respectively. All variables are defined in Appendix A.

We used the variable *DA*, which is the Manipulative Accrued Profits, to quantify information transparency; the higher the manipulative accrued profit, the lower information transparency. The results are shown in column (2) of Table 11; the coefficient of *Treat × Post* is significantly negative, which means the CSISC shareholding improves the quality and transparency of information disclosure.

Combining the results of the above regression, they support our argument that board dissenting and manipulative accrued profits are the possible intermediary channels of the positive relationship that causes the CSISC shareholding to increase firms' supervision capability and information transparency, resulting in increased ESG performance.

## 6. Conclusions

To improve investor protection, the China Securities Regulatory Commission (CSRC) aims to exercise various shareholder rights to protect minority shareholders' benefits. Based on DID analysis, we found that after the CSISC shareholding pilot reform, the ESG score in pilot regions increased by 0.091 on average, which confirmed our hypothesis that the pilot reform of CSISC shareholding has a positive influence on corporate ESG score. Furthermore, we analyzed heterogeneity and cross-sectional perspectives and found that CSISC shareholding has a positive influence on large companies and companies in non-high-polluting industries. Analysts' attention, external auditing quality, institutional shareholding, and highly-developed market intermediary and legal system can strengthen the effect of CSISC shareholding on corporate ESG performance. On the basis of previous research [5,7,17], although CSISC is a minority shareholder for companies, its policy and regulatory characteristics make its role different from that of ordinary minority shareholders, which is of special significance for corporate governance and investor protection. Hence, our paper determined a positive relationship between the CSISC and the corporate ESG rating.

The paper affirms the deterrent and supervisory governance effects of CSISC, as well as the leading demonstration effect on minority shareholders. Our findings can inspire regulators in emerging markets to establish suitable mechanisms to protect minority shareholder rights in the long run. From a policy perspective, this paper suggests that CSISC can join other forces with external governance functions, such as auditors, financial analysts, and institutional investors, activating the awareness of minority shareholders themselves to defend their rights and actively guiding and providing more convenience for minority shareholders to exercise their rights. Then, the CSISC can improve the governance and information disclosure environment of listed companies and promote the healthy development of the capital market.

There are several limitations to this study. First, because capital market policy changes frequently in China, we used DID to test our research question and conducted a series of robustness tests; however, we cannot rule out potential endogeneity concerns. Second, we mainly focused on the influence of the CSISC in pilot areas, lacking data on specific actions taken by the CSISC. We hope future studies dig deep into this issue with more data and can

also focus on the combined effect of the CSISC and other forces, such as auditors, financial analysts, and institutional investors. Moreover, because not all listed companies in China can provide accurate and detailed ESG information, we hope that with the establishment and improvement of the ESG index system, more samples can be obtained to enrich the research results, and we hope more interesting results can be found in ESG performance.

**Author Contributions:** Conceptualization, D.S. and C.X.; methodology, C.X.; software, C.X.; validation, D.S., Z.F. and C.Y.; formal analysis, C.X.; writing—original draft preparation, D.S. and Z.F.; writing—review and editing, D.S. and C.Y. All authors have read and agreed to the published version of the manuscript.

**Funding:** This research was funded by [National Social Science Fund Youth Program] grant number [22CGL010], [China Postdoctoral Science Foundation] grant number [2021M701194], [Fundamental Research Funds for the Central Universities] grant number [JKN02222203] and [Shanghai Soft Science Projects] grant number [23692121000].

**Data Availability Statement:** The datasets used and analyzed during the current study are available from the corresponding author on reasonable request.

**Conflicts of Interest:** The authors declare no conflict of interest.

## Appendix A

| Variable Name | Definition |
| --- | --- |
| *ESG* | Huazheng ESG rating |
| *Treat* | A dummy variable indicating whether the firm is in the pilot region if the company is registered in Shanghai, Guangdong excluding Shenzhen and Hunan, Treat equals one, if the company is registered in Zhejiang, Shenzhen, and Hubei, Treat equals zero. |
| *Post* | A dummy variable indicating whether the conduction of the policy, if the observation is in the year 2016 and 2017, Post equals to one, otherwise, equals to zero. |
| *Treat × Post* | The main independent variable reflects the effect of CSISC shareholding. |
| *Size* | Natural logarithm of the total assets. |
| *Lev* | The ratio of total debt to total assets. |
| *ROA* | The ratio of operating profit to total assets |
| *PB* | Market value / Total assets |
| *TobinQ* | (Net assets per share × number of non-tradable shares + price per share × number of tradable shares + book value of liabilities) / Total assets |
| *Boardsize* | Total number of the board of directors of the company |
| *Indep* | The ratio of independent directors to the board of directors |
| *Duality* | A dummy variable for CEO duality, which equals to one if a firm's CEO is also the chair of the board, and zero otherwise |
| *Top10* | The sum of squares of the shareholding ratio of top ten shareholders |
| *Loss* | The dummy variable indicates whether the company gains profit, Loss equals to one if the net profit of the current year is negative, otherwise, equals to zero. |
| *Age* | Ln (year–listing year + 1) |
| *Pre_1* | The multiplication of dummy whether the sample is in 2015 and treat. |
| *Current* | The multiplication of dummy whether the sample is in 2016 and treat |
| *Post_1* | The multiplication of dummy whether the sample is in 2017 and treat |
| *FPost* | A dummy variable indicating counterfactual policy time, if the observation is in 2012 and 2013, *FPost* equals to one, if the observation is in 2009–2011, the *FPost* equals to zero. |
| *Treat2* | A dummy variable indicating whether the firm is in the pilot region if the company is registered in Shanghai, Guangdong excluding Shenzhen and Hunan, Treat equals to one, if the company is registered in Jiangsu, Zhejiang, Shenzhen, and Hubei, Treat equals to zero. |

| Variable Name | Definition |
|---|---|
| *Datt* | A dummy variable indicates whether the firm gets high attention from analysts, if the number of research reports of the company exceeds the industry median, Datt equals to one, otherwise, equals to zero. |
| *Big4* | A dummy variable indicates whether the firm's annual report is audited by the "big four" accountancy firms if the report is audited by the big four, big4 equals to one, otherwise, equals to zero. |
| *DRIns* | A dummy variable indicates whether the company has a high institutional shareholding ratio, if the shareholding ratio of the company's institutional investors is greater than the sample median, DRIns equals to one, otherwise, equals to zero. |
| *Index* | Market Intermediary Index |
| *Vote* | The number of dissenting voting at the Annual Board of Directors. |
| *DA* | Manipulative Accrued Profits |

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
