# Peer review of "How Does a Regulatory Minority Shareholder Influence the ESG Performance? A Quasi-Natural Experiment"

_sustainability, doi:10.3390/su15076277_

Round 1

Reviewer 1 Report

The paper looks at interesting and current topic. It is well written and reads well. However it requires some improvement to bring it to publication level in an esteem journal like Sustainability.

First, the assumption or argument on how minority shareholder influence on ESG is not clear or strong, particularly in the introduction. I suggest the authors articulate the main channels through which minority shareholders affect sustainability and ESG in particular. they should include both potential and negative relationship.

The literature section provides good information on the topic but the hypothesis is not well developed. Again the potential arguments why minority shareholder could influence ESG is not clear. The argument that minority may influence ESG because of the long-term benefit is at the minimum very weak or unlikely. Arguably minority shareholders are less likely to invest in firms for long term compared with majority shareholders. Minority shareholders have less investment in the company than majority shareholders, so why should they care about the long term benefit. I think this argument should rather be use to show negative effect.

I suggest the authors provide both positive and negative arguments in the hypothesis.

In the method section, the authors should provide more explanation on how the control variables relates to ESG

The results is very comprehensive and includes many robustness. However is more of presentation of statistical tables than discussion of results. I suggest the authors provide more discussion, linking the results to hypothesis and prior studies, particular on the main results.

The conclusion should include implications of the study.

 ALL THE BEST

Author Response

Thank you for your comments concerning our manuscript. Those comments are valuable and very helpful. We have read through the comments carefully and have made corrections. Please see details below.

The paper looks at interesting and current topic. It is well written and reads well. However it requires some improvement to bring it to publication level in an esteem journal like Sustainability.

First, the assumption or argument on how minority shareholder influence on ESG is not clear or strong, particularly in the introduction. I suggest the authors articulate the main channels through which minority shareholders affect sustainability and ESG in particular. they should include both potential and negative relationship.

The literature section provides good information on the topic but the hypothesis is not well developed. Again the potential arguments why minority shareholder could influence ESG is not clear. The argument that minority may influence ESG because of the long-term benefit is at the minimum very weak or unlikely. Arguably minority shareholders are less likely to invest in firms for long term compared with majority shareholders. Minority shareholders have less investment in the company than majority shareholders, so why should they care about the long-term benefit. I think this argument should rather be use to show negative effect.

I suggest the authors provide both positive and negative arguments in the hypothesis.

(1) Thanks for the valuable suggestion.

Our paper focuses on how the China Securities Regulatory Commission (CSRC) as a new minority investor protection mechanism influence corporate ESG performance. We didn’t study the relationship between minority shareholders and ESG performance. So, as a strong investor protection mechanism, CSISC has a positive influence on ESG performance of listed companies.

CSRC is a government agency. To improve investor protection, China Securities Investor Services Center (CSISC) was established by China Securities Regulatory Commission (CSRC), which aims to exercise various shareholder rights to protect minority shareholders’ benefits. CSRC plays the role of investor protection mainly by holding100 ordinary shares of listed company and becoming a minority shareholder of the company. The rights protection services of the CSISC mainly include special representative litigation, support litigation and shareholder litigation. (The details of Institutional background can be found in section 2: CSISC shareholding.)

We find that as an effective investor protection mechanism, the pilot reform of CSISC shareholding has a positive influence on ESG performance of listed companies. The CSISC as a minority shareholder leads to a positive influence on ESG through increasing supervision capability and information transparency. We have detailed the research logic. Please refer to “1. Introduction” part in the manuscript.

In order to highlight the research question and logic of this paper, we also add a research framework in the introduction section.

In the method section, the authors should provide more explanation on how the control variables relates to ESG

(2) Thanks for the valuable suggestion.

Following Bolourian et al. (2021) and Aluchna et al. (2022), we control the variables for firm financial characteristics and governance structure which will influence the environment, social and governance (ESG) disclosure. Bolourian et al. (2021) and Aluchna et al. (2022) investigates the environment, social and governance (ESG) from perspectives of institutional investors and board structure, which similar to our paper. Therefore, we following the above papers to choose our control variables.

Please refer to “4.2 Research Methodology” in the manuscript.

Reference:

Aluchna, M., Roszkowska-Menkes, M., KamiÅ„ski, B., & Bosek-Rak, D. (2022). Do institutional investors encourage firm to social disclosure? The stakeholder salience perspective. Journal of Business Research, 142, 674-682. https://doi.org/10.1016/j.jclepro.2020.125752

The results is very comprehensive and includes many robustness. However is more of presentation of statistical tables than discussion of results. I suggest the authors provide more discussion, linking the results to hypothesis and prior studies, particular on the main results.

(3) Thanks for the valuable suggestion.

We add more discussion, linking the results to hypothesis and prior studies, on the main results and other empirical results.

Please refer to “5. Empirical results” in the manuscript.

The conclusion should include implications of the study.

(4) Thanks for the valuable suggestion.

We have added implications of the study, please refer to “6. Conclusion” part in the manuscript.

Reviewer 2 Report

The paper provides valuable information supported by data. The overall level of the paper is good. However, I have just a few small comments on the manuscript.

Captions for tables should be presented with a more specific description rather than a general sentence.

A framework or flowchart for the study is recommended for allowing a full understanding of the method and the proposed validation. This helps the reader to clearly understand the contribution of the study. Also, this would strengthen the discussion and the conclusions of the paper.

Please give more information on the data for increasing the availability and reliability of the study and understanding the results.

Conclusions need to be strengthened. Please, improve it, add some key results supported with data, and make it more effective.

Based on the key findings and conclusions, it may be better to give more recommendations for future analyses.

Author Response

The paper provides valuable information supported by data. The overall level of the paper is good. However, I have just a few small comments on the manuscript.

Captions for tables should be presented with a more specific description rather than a general sentence.

(1) Thanks for the valuable suggestion.

We add more specific description to the captions for tables which help readers clearly understand the content of the tables.

Please refer to table 3 and table 7-table9 in the manuscript.

A framework or flowchart for the study is recommended for allowing a full understanding of the method and the proposed validation. This helps the reader to clearly understand the contribution of the study. Also, this would strengthen the discussion and the conclusions of the paper.

(2) Thanks for the valuable suggestion.

Following your suggestions, we add a research framework (Figure 1) in “Introduction” of the manuscript which will strengthen the discussion and the conclusions of the paper.

Figure 1 Research Framework

Please give more information on the data for increasing the availability and reliability of the study and understanding the results.

(3) Thanks for the valuable suggestion.

Our paper uses the data of CSISC reform and ESG performance. CSISC carried out pilot projects in Shanghai, Guangdong (excluding Shenzhen) and Hunan province which started in 2016. Following the existing studies (Ge et al., 2022; Chen et al., 2023), we select samples of listed firm registered in Shanghai, Zhejiang, Guangdong, Hunan and Hubei provinces from 2013 to 2017. The data of ESG performance froms Huazheng ESG, and lots of paper use this data to study this topic (Chen and Xie, 2022; Jiang et al., 2022; Deng et al., 2023).

Please refer to “4.1 Sample and Data” in the manuscript.

Conclusions need to be strengthened. Please, improve it, add some key results supported with data, and make it more effective.

(4) Thanks for the valuable suggestion.

We have improved the conclusions, please refer to “6. Conclusion” part in the manuscript.

Based on the key findings and conclusions, it may be better to give more recommendations for future analyses.

(5) Thanks for the valuable suggestion.

We have given some recommendations for future analyses, please refer to “6. Conclusion” part in the manuscript.

Reviewer 3 Report

Title:  How does Regulatory Minority Shareholder influence ESG performance? A quasi-natural experiment

After reviewing this article, I think it is potential for publication but the authors should revise as comments below:

 - The introduction is well presented, however, the authors should emphasize the research question clearly. I still don't see clearly what the research question is.

- In the literature review, the author must discuss the theories relating to your topic that can support your hypothesis. Moreover, the authors should review and update recent studies relating to environmental, social, and governance. I suggest the authors review and cite some recent studies such as Dang and Nguyen (2021); Landi, Iandolo et al. (2022); Nguyen (2022); Nguyen and Dang (2022); Nguyen (2022)… (see reference).

- In section 4.2, why did you not control the industry effect? In this section, you must explain the estimation method applied in this study including robustness tests.

- The use of control variables must be explained more instead of stating that you are based on the literature.

-  There are some typos and grammatical errors, you must recheck it carefully.

References

Dang, V. C. and Q. K. Nguyen (2021). "Internal corporate governance and stock price crash risk: evidence from Vietnam." Journal of Sustainable Finance & Investment: 1-18.

Landi, G. C., et al. (2022). "Embedding sustainability in risk management: The impact of environmental, social, and governance ratings on corporate financial risk." Corporate Social Responsibility and Environmental Management 29(4): 1096-1107.

Nguyen, Q. K. (2022). "Determinants of bank risk governance structure: A cross-country analysis." Research in International Business and Finance 60: 101575.

Nguyen, Q. K. (2022). "The impact of risk governance structure on bank risk management effectiveness: evidence from ASEAN countries." Heliyon: e11192.

Nguyen, Q. K. and V. C. Dang (2022). "Does the country’s institutional quality enhance the role of risk governance in preventing bank risk?" Applied Economics Letters: 1-4.

Author Response

After reviewing this article, I think it is potential for publication but the authors should revise as comments below:

 - The introduction is well presented, however, the authors should emphasize the research question clearly. I still don't see clearly what the research question is.

(1) Thanks for the valuable suggestion.

CSRC is a government agency. To improve investor protection, China Securities Investor Services Center (CSISC) was established by China Securities Regulatory Commission (CSRC), which aims to exercise various shareholder rights to protect minority shareholders’ benefits.

Based on the existing literature, this paper will further focus on how CSISC can improve the long-term sustainability performance of companies, which means the impact on ESG performance. We find that the pilot reform of CSISC shareholding has a positive influence on ESG performance of listed companies. The CSISC as a minority shareholder leads to a positive influence on ESG through increasing supervision capability and information transparency. We have detailed the research logic and research question. Please refer to “1. Introduction” part in the manuscript.

In order to highlight the research question and logic of this paper, we add a research framework in the introduction section.

- In the literature review, the author must discuss the theories relating to your topic that can support your hypothesis. Moreover, the authors should review and update recent studies relating to environmental, social, and governance. I suggest the authors review and cite some recent studies such as Dang and Nguyen (2021); Landi, Iandolo et al. (2022); Nguyen (2022); Nguyen and Dang (2022); Nguyen (2022)… (see reference).

(2) Thanks for the valuable suggestion.

We have added relevant literature in the paper.

- In section 4.2, why did you not control the industry effect? In this section, you must explain the estimation method applied in this study including robustness tests.

(3) Thanks for the valuable suggestion.

We perform a difference-in-difference (DID) approach and control firm fixed effects and year fixed effects in research model (1) that means there is no need to control for industry based on the principles of statistics and econometrics.

And following the existing DID approach, we perform some robustness tests including parallel trend analysis, placebo test, propensity score matching, controlling time and regional effects and changing control group to validate our DID research design and confirm our results. These robustness tests follow Ge et al.(2022) and Chen et al.(2023).

Reference:

Ge, W., Ouyang, C., Shi, Z., Chen, Z., 2022. Can a not-for-profit minority institutional shareholder make a big difference in corporate governance? A quasi-natural experiment. Journal of Corporate Finance 72, 102125. https://doi.org/10.1016/j.jcorpfin.2021.102125

Chen, S., Chen, Y., Zhang, D., Wang, J., 2023. Can minority investor activism promote corporate risk-taking? Evidence from a quasi-natural experiment in China. International Review of Financial Analysis 85, 102430. https://doi.org/10.1016/j.irfa.2022.102430

- The use of control variables must be explained more instead of stating that you are based on the literature.

(4) Thanks for the valuable suggestion.

Following Bolourian et al. (2021) and Aluchna et al. (2022), we control the variables for firm financial characteristics and governance structure which will influence the environment, social and governance (ESG) disclosure. Bolourian et al. (2021) and Aluchna et al. (2022) investigates the environment, social and governance (ESG) from perspectives of institutional investors and board structure, which similar to our paper. Therefore, we following the above papers to choose our control variables.

Please refer to “4.2 Research Methodology” in the manuscript.

Reference:

Aluchna, M., Roszkowska-Menkes, M., KamiÅ„ski, B., & Bosek-Rak, D. (2022). Do institutional investors encourage firm to social disclosure? The stakeholder salience perspective. Journal of Business Research, 142, 674-682. https://doi.org/10.1016/j.jclepro.2020.125752

-  There are some typos and grammatical errors, you must recheck it carefully.

(5) Thanks for the valuable suggestion.

We have modified typos and grammatical errors.

Reviewer 4 Report

kindly see all my comments addressed in the file attached and i do prefer to read the recent publication like: 

Treepongkaruna, S., Kyaw, K., & Jiraporn, P. (2022). Shareholder litigation rights and ESG controversies: A quasi-natural experiment. International Review of Financial Analysis84, 102396.

Author Response

Thank you for your comments concerning our manuscript. Those comments are valuable and very helpful. We have read through the comments carefully and have made corrections. Please see the details below.

  1.  

Thanks for the valuable suggestion. This content is the research conclusion of this paper, which is reached through the empirical test. Please refer to “5.4. Cross-sectional Analysis” part in the manuscript.

2.

Thanks for the valuable suggestion. The definition of all the variables please refer to “4.2. Research Methodology and Appendix A” part in the manuscript.

3.

Thanks for the valuable suggestion. We have added relevant literature in the paper.

4.

Thanks for the valuable suggestion. We have added relevant literature in the paper.

5.

Thanks for the valuable suggestion. We have added relevant literature in the paper.

6.

Thanks for the valuable suggestion. We have added relevant literature in the paper.

We have deleted the unrelated literature (Ding et al., 2022; Govindan et al., 2021) in the paper.

7.

Thanks for the valuable suggestion. We have modified the statement and added relevant literature in the paper.

8.

Thanks for the valuable suggestion. We have modified the statement and added relevant literature in the paper.

Round 2

Reviewer 3 Report

Article:  How does Regulatory Minority Shareholder influence ESG performance? A quasi-natural experiment

After reviewing this version, I still have some concerns that the authors need to consider revising as comments below:

- The authors should consider the studies I suggest in my previous comments

- The authors need to explain in detail why to control selected variables, for example, how firm size affects ESG and similarly for other control variables. The authors were unable to list multiple variables on the grounds that they were based on another study. Also, the author's citations are inconsistent. For example, the authors claim that control variables are derived from Bolourian et al (2021), but this study is "the impact of corporate governance on corporate social responsibility at the board level". CRS and ESG are similar?

- The argument that "the influence of CSISC shareholding may be different between companies of different size" needs to be specifically cited based on theory or previous studies

- There are still a lot of grammatical errors, the authors need to check carefully. For example, “investigates” not “investigate” (p.3); “exists differences” (p.16)…

n general, the authors have not yet resolved some of my concerns in the previous comments. Authors should review and edit before publication.

Author Response

After reviewing this version, I still have some concerns that the authors need to consider revising as comments below:

- The authors should consider the studies I suggest in my previous comments

- The authors need to explain in detail why to control selected variables, for example, how firm size affects ESG and similarly for other control variables. The authors were unable to list multiple variables on the grounds that they were based on another study. Also, the author's citations are inconsistent. For example, the authors claim that control variables are derived from Bolourian et al (2021), but this study is "the impact of corporate governance on corporate social responsibility at the board level". CRS and ESG are similar?

(1) Thanks for the comments.

Following your suggestions, we read some literatures on ESG. For example, Jang et al. (2022a) investigate the effect of managers’ manipulative tendencies on firms’ environmental, social, and governance (ESG) performance; Huang et al. (2022) studies how controlling shareholders pledging affect ESG; Jiang et al. (2022b) studies the relationship between institutional investors’ corporate site visits and environmental,

social, and governance (ESG) performance; Wang et al. (2023) examines whether institutional investors affect the environment, social, and governance (ESG) performance of Chinese-listed companies. These papers control corporate financial characteristics (e.g. firm size, asset-liability ratio, return on asset, market performance, profitability), governance structure (e.g. board size, board independence, CEO duality, ownership concentration), and other variables (e.g. firm age). The above control variables have an impact on ESG. For example, firms with larger scale and better growth may perform better in ESG. We need to control these variables which may influence ESG. Therefore, we follow the above papers to choose our control variables, which are common variables that must be controlled in corporate governance and corporate finance research.

Please refer to “4.2 Research Methodology” in the manuscript.

Reference:

Jang, G. Y., Kang, H. G., & Kim, W. (2022a). Corporate executives’ incentives and ESG performance. Finance Research Letters, 49, 103187. https://doi.org/10.1016/j.frl.2022.103187.

Jiang, Y., Wang, C., Li, S., & Wan, J. (2022b). Do institutional investors' corporate site visits improve ESG performance? Evidence from China. Pacific-Basin Finance Journal, 76, 101884. https://doi.org/10.1016/j.pacfin.2022.101884.

Huang, W., Luo, Y., Wang, X., & Xiao, L. (2022). Controlling shareholder pledging and corporate ESG behavior. Research in International Business and Finance, 61, 101655. https://doi.org/10.1016/j.ribaf.2022.101655.

Wang, Y., Lin, Y., Fu, X., & Chen, S. (2023). Institutional ownership heterogeneity and ESG performance: Evidence from China. Finance Research Letters, 51, 103448. https://doi.org/10.1016/j.frl.2022.103448.

- The argument that "the influence of CSISC shareholding may be different between companies of different size" needs to be specifically cited based on theory or previous studies.

(2) Thanks for the comments. Drempetic et al.(2020) show that larger firms have more

resources and more often use reporting tools to provide ESG data. They note that larger firms use more instruments to analyze and report ethical and sustainable behavior. Following this paper, we try to investigate whether the influence of CSISC shareholding may be different between companies of different sizes.

Please refer to “5.3.1. Size heterogeneity” in the manuscript.

Reference:

Drempetic, S., Klein, C., & Zwergel, B. (2020). The influence of firm size on the ESG score: Corporate sustainability ratings under review. Journal of Business Ethics, 167, 333-360. https://doi.org/10.1007/s10551-019-04164-1

- There are still a lot of grammatical errors, the authors need to check carefully. For example, “investigates” not “investigate” (p.3); “exists differences” (p.16)…

(2) Thanks for the comments. We have read the full text and carefully revised the wrong content.

in general, the authors have not yet resolved some of my concerns in the previous comments. Authors should review and edit before publication.

Reviewer 4 Report

 all comments have been considered. 

Round 3

Reviewer 3 Report

After looking at the edit I see that this is the 2nd edit but the authors still don't explain control variables. The authors say that the authors based on previous studies such as Bolourian et al. (2021) or  Jang et al (2022a, b), however, Bolourian et al. (2021) does not study ESG but CRS. You are based on Jang et al (2022a, b) but do not control corporate executives’ incentives and institutional investors' corporate site visits. Therefore, the lack of important control variables can render the model meaningless. In addition, the lack of clarification in the article will not prove that the model is reliable.

Author Response

Thank you for your comments concerning our manuscript. Those comments are valuable and very helpful. We have read through the comments carefully and have made corrections. Please see details below.

After reviewing this version, I still have some concerns that the authors need to consider revising as comments below:

- The authors should consider the studies I suggest in my previous comments

- The authors need to explain in detail why to control selected variables, for example, how firm size affects ESG and similarly for other control variables. The authors were unable to list multiple variables on the grounds that they were based on another study. Also, the author's citations are inconsistent. For example, the authors claim that control variables are derived from Bolourian et al (2021), but this study is "the impact of corporate governance on corporate social responsibility at the board level". CRS and ESG are similar?

(1) Thanks for the comments.

Following your suggestions, we read some literatures on ESG. For example, Jang et al. (2022a) investigate the effect of managers’ manipulative tendencies on firms’ environmental, social, and governance (ESG) performance; Huang et al. (2022) studies how controlling shareholders pledging affect ESG; Jiang et al. (2022b) studies the relationship between institutional investors’ corporate site visits and environmental,

social, and governance (ESG) performance; Wang et al. (2023) examines whether institutional investors affect the environment, social, and governance (ESG) performance of Chinese-listed companies. These papers control corporate financial characteristics (e.g. firm size, asset-liability ratio, return on asset, market performance, profitability), governance structure (e.g. board size, board independence, CEO duality, ownership concentration), and other variables (e.g. firm age). The above control variables have an impact on ESG. For example, firms with larger scale and better growth may perform better in ESG. We need to control these variables which may influence ESG. Therefore, we follow the above papers to choose our control variables, which are common variables that must be controlled in corporate governance and corporate finance research.

Please refer to “4.2 Research Methodology” in the manuscript.

Reference:

Jang, G. Y., Kang, H. G., & Kim, W. (2022a). Corporate executives’ incentives and ESG performance. Finance Research Letters, 49, 103187. https://doi.org/10.1016/j.frl.2022.103187.

Jiang, Y., Wang, C., Li, S., & Wan, J. (2022b). Do institutional investors' corporate site visits improve ESG performance? Evidence from China. Pacific-Basin Finance Journal, 76, 101884. https://doi.org/10.1016/j.pacfin.2022.101884.

Huang, W., Luo, Y., Wang, X., & Xiao, L. (2022). Controlling shareholder pledging and corporate ESG behavior. Research in International Business and Finance, 61, 101655. https://doi.org/10.1016/j.ribaf.2022.101655.

Wang, Y., Lin, Y., Fu, X., & Chen, S. (2023). Institutional ownership heterogeneity and ESG performance: Evidence from China. Finance Research Letters, 51, 103448. https://doi.org/10.1016/j.frl.2022.103448.

- The argument that "the influence of CSISC shareholding may be different between companies of different size" needs to be specifically cited based on theory or previous studies.

(2) Thanks for the comments. Drempetic et al.(2020) show that larger firms have more

resources and more often use reporting tools to provide ESG data. They note that larger firms use more instruments to analyze and report ethical and sustainable behavior. Following this paper, we try to investigate whether the influence of CSISC shareholding may be different between companies of different sizes.

Please refer to “5.3.1. Size heterogeneity” in the manuscript.

Reference:

Drempetic, S., Klein, C., & Zwergel, B. (2020). The influence of firm size on the ESG score: Corporate sustainability ratings under review. Journal of Business Ethics, 167, 333-360. https://doi.org/10.1007/s10551-019-04164-1

- There are still a lot of grammatical errors, the authors need to check carefully. For example, “investigates” not “investigate” (p.3); “exists differences” (p.16)…

(2) Thanks for the comments. We have read the full text and carefully revised the wrong content.

in general, the authors have not yet resolved some of my concerns in the previous comments. Authors should review and edit before publication.